# A phase I open-label clinical trial to study drug-drug interactions of Dorzagliatin and Sitagliptin in patients with type 2 diabetes and obesity

Li Chen [1] ✉, Jiayi Zhang [1], Yu Sun [1], Yu Zhao [1], Xiang Liu [1], Zhiyin Fang [1], Lingge Feng [1], Bin He[1], Quanfei Zou[1] & Gregory J. Tracey[2]

This is a phase 1, open-label, single-sequence, multiple-dose, single-center trial conducted in the US (NCT03790839), to evaluate the clinical pharmacokinetics, safety and pharmacodynamics of dorzagliatin co-administered with sitagliptin in patients with T2D and obesity. The trial has completed. 15 patients with T2D and obesity were recruited and treated with sitagliptin 100 mg QD on Day 1-5, followed by a combination of sitagliptin 100 mg QD with dorzagliatin 75 mg BID at second stage on Day 6-10 and the third stage of dorzagliatin 75 mg BID alone on Day 11-15. Primary outcomes include pharmacokinetic geometric mean ratio (GMR), safety and tolerability. Secondary outcomes include the incremental area under the curve for 4 hours post oral glucose tolerance test (iAUC) of pharmacodynamic biomarkers and glucose sensitivity. GMR for $AUC_{0-24h}$ and $C_{max}$ were 92.63 (90% CI, 85.61, 100.22) and 98.14 (90% CI, 83.73, 115.03) in combination/sitagliptin, and 100.34 (90% CI, 96.08, 104.79) and 102.34 (90% CI, 86.92, 120.50) in combination/dorzagliatin, respectively. Combination treatment did not increase the adverse events and well-tolerated in T2D patients. Lack of clinically meaningful pharmacokinetic interactions between dorzagliatin and sitagliptin, and an improvement of glycemic control under combination potentially support their co-administration for diabetes management.

Glucokinase (GK) is a glucose sensor and plays a central role in glucose homeostasis[1]. It is expressed in the endocrine organs of the pancreas alpha and beta cells to regulate glucagon and insulin secretion, and in intestinal L cells to regulate glucagon-like peptide-1 (GLP-1) secretion, as a metabolic sensor to convert the signal of glucose metabolism into glucose threshold-controlled hormone release. The majority of GK is expressed in the liver to convert the post-meal glucose into glycogen mediated by insulin and glucagon. GLP-1 and glucagon-like peptide-1 receptor agonist (GLP-1RA) have been studied intensively for the treatment of diabetes and obesity through peripheral and central

receptors[2–5]. Current therapies have been evolved into a dual active peptide of GLP-1 and glucose-dependent insulinotropic peptide (GIP)[6,7]. GLP-1 originated from intestinal L cell and pancreatic alpha-cell, together with its circulating metabolites generated by dipeptidyl peptidase 4 (DPP-4) enzyme, were considered to play an important role in glucose homeostasis through glucagon-like peptide-1 receptor (GLP-1R), as well as having cardiovascular and neurological protection effects through non-GLP-1R regulated functions[8–10]. Therefore, the strategy of developing GLP-1R-based therapy may have limited the physiological function of incretins and its metabolites beyond their

[1]Hua Medicine (Shanghai) Limited, Shanghai, China. [2]Frontage Clinical Services, Inc., Secaucus, NJ, USA. ✉e-mail: lichen@huamedicine.com

role in glucose homeostasis. Defects of GLP-1 secretion in response to glucose challenge have been reported in western and eastern patients with impaired glucose tolerance (IGT) and type 2 diabetes (T2D)[11,12], which can be rescued by either GLP-1RA, or ideally a glucose-dependent regulator on GLP-1 secretion. Dorzagliatin is an orally active allosteric glucokinase activator (GKA) which acts on GK in the pancreas and liver for the treatment of T2D[13–16]. As a new class of diabetes medicine, dorzagliatin was considered safe and effective in T2D patients for glycemic control and may have a unique advantage in the treatment of diabetic kidney disease patients, given its minimal excretion from the kidney[17,18]. It has shown that dorzagliatin improves glycemic control and increases early-phase insulin secretion in T2D patients in Chinese T2D patients after 1 month of treatment[13]. We have also observed a sustained effect in the improvement of disposition index and HOMA-IR one week after drug withdrawal in a 3-month monotherapy study in which dorzagliatin showed a dose-dependent glycated hemoglobin (HbA1c) reduction of 1.2% in a 75 mg BID group[14]. In the two phase 3 studies, T2D subjects received 75 mg dorzagliatin twice a day either alone in the drug naïve patients (SEED Study) or combined with metformin in the metformin-tolerated patients (DAWN Study) for 6 months in the double-blinded, placebo-controlled and randomized study followed by an open-label 28-week extension with dorzagliatin for its safety evaluation. Dorzagliatin is safe and well tolerated with an average HbA1c reduction of 1% from baseline with a 42–44% glycemic control rate in both trials with minimum hypoglycemia[15,16]. A logistic regression study showed that the improvement of early-phase insulin secretion measured by insulinogenic index and disposition index is the major factor for achieving glycemic control[19]. Additional clinical studies to investigate its mechanism of action include a double-blinded crossover trial in glucokinase–maturity-onset diabetes of the young (GCK-MODY) patients who suffered from a heterozygous GK gene inactive mutation and defect in second phase insulin secretion in response to a glucose challenge. Dorzagliatin improves the second phase insulin secretion and glucose sensitivity in this study[20].

DPP-4 is an enzyme that converts the active GLP-1 and GIP peptide hormone into its inactive metabolite GLP-1 (9-36) amide (GLP-1m), and thus a therapeutic target for diabetes. Sitagliptin is a first-in-class DPP-4 inhibitor launched in 2006 for T2D through its effect to increase incretin levels of GLP-1 and GIP, which increase insulin secretion and decrease gastric emptying. The combination of sitagliptin with other oral antidiabetic drugs (OADs) in glycemic control has become a common clinical practice.

Here, we show there is a lack of clinically meaningful pharmacokinetic interactions between dorzagliatin and sitagliptin. Dorzagliatin regulates glucose-stimulated GLP-1 release and improves glycemic control with good tolerance when combined with sitagliptin in patients with T2D and obesity, suggesting the role of dorzagliatin in the regulation of GLP-1 release in response to oral glucose challenge in a triple acting role of regulation of glucose homeostasis.

## Results
### Study population
The subject demographics and baseline characteristics are summarized in Table 1.

A total of 15 subjects were enrolled and included in pharmacodynamics (PD) and safety analysis. Fourteen subjects were included in the pharmacokinetics (PK) analysis, as one subject discontinued due to an adverse event (AE) (an erythematous rash) on Day 11. The cohort had an average age (mean ± SD) of 56.70 ± 5.39 years, body mass index (BMI) of 32.06 ± 3.57 kg/m², HbA1c 8.24 ± 0.99%, and fasting blood glucose 178.90 ± 44.79 mg/dL.

### PK
PK parameters maximum plasma concentration ($C_{max}$) and area under the concentration-time curve from 0 to 24 h ($AUC_{0–24h}$) for sitagliptin

**Table 1 | Demographic characteristics of study subjects**

|  | Subjects N = 15 |
|---|---|
| Age (yrs) | 56.70 (5.39) |
| Gender, n (%) |  |
| Male | 4 (26.67) Age 40–64 |
| Female | 11 (73.33) Age 54–62 |
| BMI (kg/m²) | 32.06 (3.57) |
| Race |  |
| Black/African American | 3 (20.00) |
| White | 12 (80.00) |
| Ethnicity, n (%) |  |
| Hispanic/Latino | 14 (93.33) |
| Not Hispanic/Latino | 1 (6.67) |
| HbA1c (%) | 8.24 (0.99) |
| FBG (mg/dL) | 178.90 (44.79) |

Values are presented as the mean (SD) or n (%).
BMI body mass index, FBG fasting blood glucose, HbA1c glycated hemoglobin.

were similar when sitagliptin was administered alone or co-administered with dorzagliatin, which are illustrated in Table 2. The corresponding adjusted geometric mean ratio (GMR) (combination/monotherapy ratio) of sitagliptin were 92.63 (90% confidence interval (CI), 85.61, and 100.22) for $AUC_{0–24h}$ and 98.14 (90% CI, 83.73, 115.03) for $C_{max}$, respectively.

PK parameters $C_{max}$ and $AUC_{0–24h}$ for dorzagliatin were similar when dorzagliatin was administered alone and co-administered with sitagliptin, which are illustrated in Table 2. The GMR of dorzagliatin were 100.34 (90% CI, 96.08, 104.79) for $AUC_{0–24h}$ and 102.34 (90% CI, 86.92, 120.50) for $C_{max}$, respectively.

The GMR and 90% CIs for the $AUC_{0–24h}$ and $C_{max}$ for sitagliptin or dorzagliatin fell within the standard bioequivalence boundaries of 80–125%[21] in Supplementary Fig. S2, indicating that co-administration of sitagliptin and dorzagliatin did not significantly affect the PK of either of them.

### Safety and tolerability
Safety and tolerability were assessed by reviewing individual data from all the enrolled subjects in the study. No clinically significant findings or overall changes was seen in clinical laboratory tests, vital signs, physical examination, or standard 12-lead electrocardiograms (ECGs) examinations, and no deaths or drug-related serious treatment-emergent adverse events (TEAEs) occurred in the study (Supplementary Table S1). All TEAEs were in mild or moderate severity and mostly unrelated to study drugs.

There is no increased frequency or severity of any AE during the combination treatment of sitagliptin and dorzagliatin observed, compared with either monotherapy of sitagliptin or dorzagliatin. One case of hypoglycemia was reported during the combination treatment (sitagliptin + dorzagliatin) in mild severity and relieved quickly without intervention, which was assessed possibly due to an inadequate/untimely dietary intake as it occurred very close to lunch time on the testing day when a glucose solution was given in lieu of a regular breakfast. Thus, the combination of dorzagliatin with sitagliptin did not increase the risk of severe hypoglycemia in the safety profiles. Multiple doses of dorzagliatin alone (75 mg BID), or in combination with sitagliptin (100 mg QD) were safe and well-tolerated.

### PD
A clear result was observed on the glucose-lowering effect with combination treatment compared with sitagliptin or dorzagliatin monotherapy. The incremental area under the curve for 4 h ($iAUC_{0–4h}$) of

**Table 2 | Pharmacokinetic parameters for sitagliptin and dorzagliatin, measured at steady state, after the last dose in a 5-day interval of sitagliptin only, dorzagliatin only, or sitagliptin + dorzagliatin treatment**

| | Geometric mean | | | GMR (90% Cl) |
|---|---|---|---|---|
| | Sitagliptin alone | Sitagliptin + Dorzagliatin | Dorzagliatin alone | |
| Sitagliptin | | | | |
| $C_{max}$ (ng/mL) | 410 | 403 | / | 98.14 (83.73–115.03) |
| $AUC_{0-24h}$ (ng*h/mL) | 2938 | 2722 | / | 92.63 (85.61–100.22) |
| Dorzagliatin | | | | |
| $C_{max}$ (ng/mL) | / | 833 | 814 | 102.34 (86.92–120.50) |
| $AUC_{0-24h}$ (ng*h/mL) | / | 6593 | 6571 | 100.34 (96.08–104.79) |

$C_{max}$ maximum plasma concentration, $AUC_{0-24}$ area under the concentration-time curve from 0 to 24 h, GMR adjusted geometric mean ratios (combination/monotherapy ratio), CI confidence interval.

glucose in oral glucose tolerance test (OGTT) under sitagliptin alone, sitagliptin+dorzagliatin and dorzagliatin alone were 378.00 ± 87.80, 253.00 ± 116.00, and 339.00 ± 124.00 mg × h/dL, respectively (Fig. 1). Similar trends were observed in the incremental maximum concentration from fasting state (at time of 0) before OGTT ($iC_{max}$) of glucose with a reduction to 142 mg/dL in combination from 165 mg/dL in sitagliptin alone ($p < 0.05$). The effect of improved glycemic control was associated with a significant increase of C-peptide $iC_{max}$ from 5.10 ng/mL in sitagliptin to 6.33 ng/mL in combination ($p < 0.05$), as shown in Table 3. The mean serum glucose concentration-time curve is illustrated in Supplementary Fig. S3a.

These results manifest the combination treatment of dorzagliatin with sitagliptin achieved a greater glucose-lowering effect than sitagliptin or dorzagliatin monotherapy.

Consistent with glucose outcomes, combination treatment showed higher C-peptide secretion compared with sitagliptin or dorzagliatin monotherapy under glucose stimulation, as shown in Fig. 1.

The $iAUC_{0-4h}$ of C-peptide in OGTT under sitagliptin alone, sitagliptin+dorzagliatin, and dorzagliatin alone were 12.10 ± 5.35, 14.40 ± 7.18, and 9.82 ± 3.86 ng × h/mL, respectively. Similar trends were also observed in $iC_{max}$, which are summarized in Table 3 that the incremental maximum C-peptide level was observed in the combination treatment over the monotherapy significantly ($p < 0.05$ with sitagliptin and $P < 0.01$ with dorzagliatin). The serum C-peptide concentration-time curve is illustrated in Supplementary Fig. S3b.

These results manifest the combination treatment of dorzagliatin with sitagliptin improved glucose-stimulated insulin secretion (GSIS) to achieve a greater glucose-lowering effect than either monotherapy, thus indicating the synergic potential for glycemic control in T2D patients.

GLP-1 was measured in total and active forms. Interestingly, for GLP-1 secretion in the OGTT, dorzagliatin monotherapy resulted in a significantly higher $GLP-1_{total}$ level compared with combination treatment. The $iAUC_{0-4h}$ of $GLP-1_{total}$ in OGTT under sitagliptin alone, sitagliptin + dorzagliatin, and dorzagliatin alone were 14.70 ± 15.50, 11.20 ± 9.85, and 22.80 ± 11.70 pmol × h/L, respectively (Fig. 1), with similar trends in $iC_{max}$ as shown in Table 3, in which dorzagliatin resulted in the highest level of glucose-stimulated GLP-1 release of 22.10 pmol/L compared with the combination therapy of 9.17 pmol/L in $iC_{max}$ ($p < 0.05$). The plasma $GLP-1_{total}$ concentration-time curve is illustrated in Supplementary Fig. S3c, showing the time to reach maximum plasma concentration ($T_{max}$) is 30 min after glucose challenge or 1 h after drug administration. Correlation of PK-PD analysis showed that dorzagliatin drug concentration maintained above 500 ng/mL from 30 min post-drug dosing to 4 h, where total GLP-1 secretion mainly occurred from 30 min to 2 h with $T_{max}$ at 1 h (Supplementary Fig. S4), suggesting a dorzagliatin regulated glucose-stimulated GLP-1 secretion.

Combination treatment obtained numerically increased $GLP-1_{active}$ compared with sitagliptin or dorzagliatin monotherapy (Fig. 1 and Supplementary Fig. S3d). The $iAUC_{0-4h}$ of $GLP-1_{active}$ in OGTT under sitagliptin alone, sitagliptin + dorzagliatin, and dorzagliatin alone were 10.80 ± 5.93, 12.20 ± 8.21, and 6.26 ± 3.73 pmol × h/L, respectively (Fig. 1), with similar trends in $iC_{max}$ (Table 3), in which sitagliptin combination with dorzagliatin resulted in the highest level of circulating $GLP-1_{active}$. The plasma $GLP-1_{active}$ concentration-time curve is illustrated in Supplementary Fig. S3d.

The serum C-peptide and plasma GLP-1 response to glucose under the three treatment regimens post-oral glucose challenge are illustrated in Fig. 2, and glucose sensitivity comparisons are detailed in Table 3.

Combination treatment obtained significantly increased early insulinogenic index $\Delta C_{30}/\Delta G_{30}$ compared with sitagliptin or dorzagliatin monotherapy, with 0.04 ± 0.04 ng/mL per mg/dL, 0.02 ± 0.01 ng/mL per mg/dL, and 0.02 ± 0.01 ng/mL per mg/dL, respectively ($p < 0.05$). Similar trends were observed in GLP-1 secretion index $\Delta GLP-1_{active.30}/\Delta G_{30}$, with numerically higher $GLP-1_{active}$ response to glucose in combination treatment compared with either monotherapy, whereas the total GLP-1 secretion index is significantly higher in dorzagliatin monotherapy with 0.23 ± 0.14 pmol/L per mg/dL, and 0.12 ± 0.18 pmol/L per mg/dL in sitagliptin monotherapy and 0.11 ± 0.10 pmol/L per mg/dL in combination therapy ($p < 0.05$), respectively.

## Discussion

Dorzagliatin demonstrated additional benefits in blood glucose reduction when it is combined with sitagliptin. Although both drugs act on the decrease of postprandial glucose, an additional 30% reduction of glucose is observed in the OGTT when dorzagliatin was added to sitagliptin over either monotherapy ($p < 0.05$). The benefit in glycemic control is correlated with an increased GSIS in $iAUC_{0-4h}$ and improvement of early-phase C-peptide secretion index ($\Delta C_{30}/\Delta G_{30}$) in the combination over each monotherapy with statistical significance. Based on the data from the current study, the plasma concentration of $GLP-1_{active}$ is increased almost onefold for the combination of dorzagliatin and sitagliptin over dorzagliatin alone, which is correlated with an increased glucose-stimulated C-peptide secretion by nearly 50%. Accordingly, the enhanced beta-cell secretion function was associated with an improved GLP-1 secretion index in combination treatment (Fig. 2 and Table 3).

The statistically significant improvement of early-phase C-peptide secretion index and the increased $GLP-1_{active}$ secretion should have positively contributed to the glucose sensitivity and glycemic control when dorzagliatin is combined with a DPP-4 inhibitor sitagliptin. It is conceivable that the combination worked together to restoration of GK function in the pancreas and intestine simultaneously, which triggered the GK-mediated GSIS in beta cells and engaged the GLP-1 secretion to further enhance the GSIS function[22]. Improvement of hepatic glucose metabolism by dorzagliatin is found in a relationship with effective reduction of fasting plasma glucose in a healthy subject without change of insulin secretion, and reduction of postprandial glucose in T2D

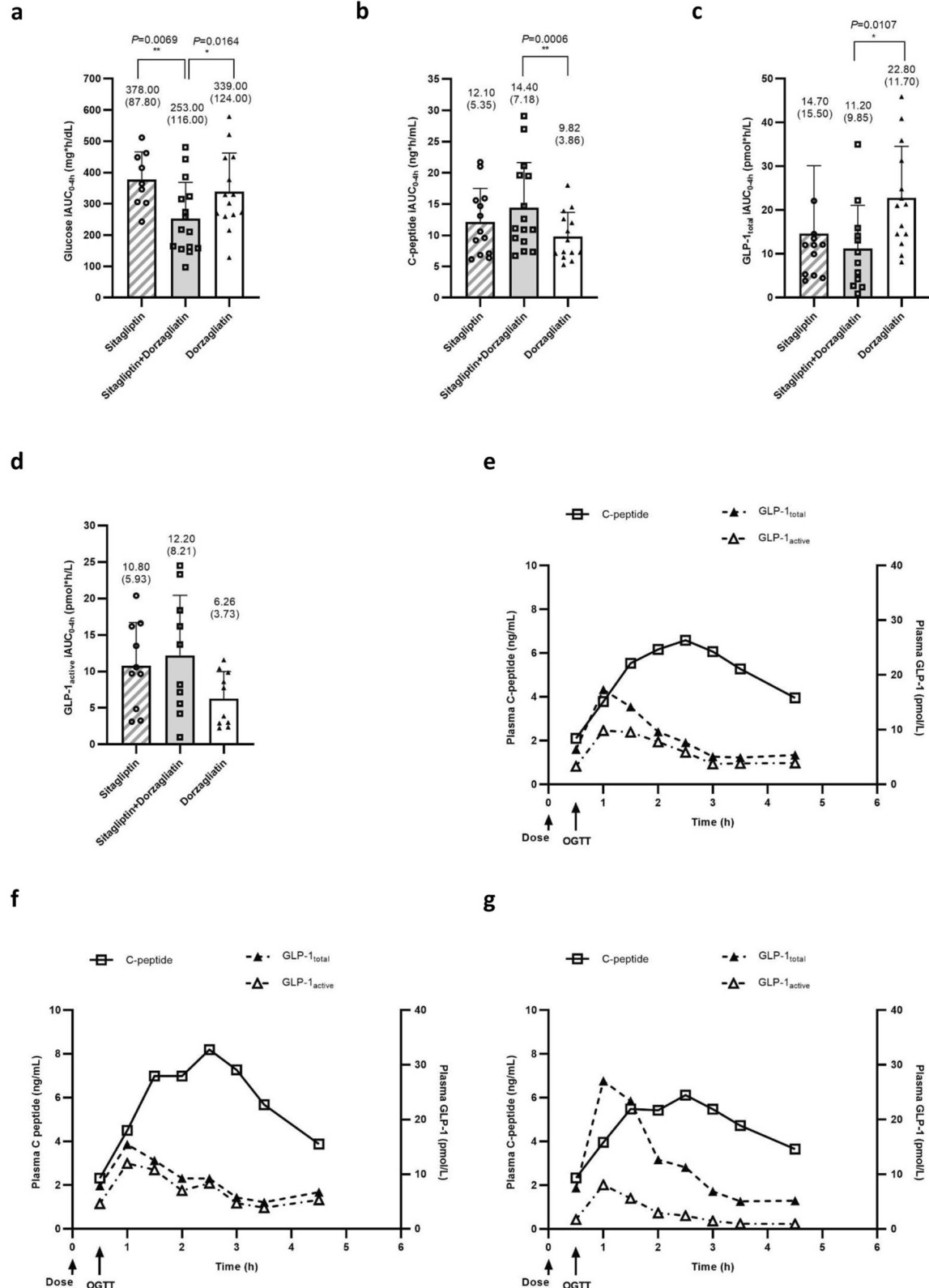

patients[13,23], and will be further studied in subsequent trials (R Basu, ClinicalTrials.gov identifier: NCT05098470). It has been reported that a substantial reduction of hepatic GK expression in T2D patients correlated with their hyperglycemia and hepatic insulin resistance[24,25]. Epigenetic modification of hepatic GK promoter caused a reduction of hepatic GK expressions and a reduced glycogen content in the liver[26]. Dorzagliatin restored the hepatic GK expression in diabetes rats,

increased the numbers of insulin-secreting cells in the pancreas, and improved glycemic control after 28-day treatment[27]. In the same animal model, dorzagliatin increased fasting GLP-1 level in the small intestine and pancreas, either alone or in combination with sitagliptin[28]. The effect on glucose-stimulated GLP-1 secretion was not reported in other GKAs and was considered a unique feature of dorzagliatin. In the preclinical studies, we discovered its effect on glucose-stimulated

**Fig. 1 | iAUC and plasma concentration of glucose, C-peptide, and GLP-1.**
iAUC$_{0-4h}$ of **a** glucose (sitagliptin, $n = 9$; sitagliptin + dorzagliatin, $n = 15$; dorzagliatin, $n = 14$), **b** C-peptide (sitagliptin, $n = 13$; sitagliptin + dorzagliatin, $n = 15$; dorzagliatin, $n = 14$), **c** GLP-1$_{total}$ (sitagliptin, $n = 12$; sitagliptin + dorzagliatin, $n = 12$; dorzagliatin, $n = 14$), and **d** GLP-1$_{active}$ (sitagliptin, $n = 10$; sitagliptin + dorzagliatin, $n = 10$; dorzagliatin, $n = 10$) in OGTT studies. Diagonal bar and empty circles represent mean and individual iAUC under sitagliptin monotherapy, gray bar and empty squares represent mean and individual iAUC under sitagliptin+dorzagliatin therapy, white bar and empty triangles represent mean and individual iAUC under dorzagliatin monotherapy. Error bars represent standard deviation; $P$ values were calculated for the comparisons between the PD parameters (log-difference) using the mixed models; Statistical tests were two-sided at a significance level of 0.05, and no adjustments were made for multiplicity. The plasma concentration-time curve of GLP-1 and C-peptide **e** under sitagliptin monotherapy, **f** under sitagliptin +dorzagliatin therapy, and **g** under dorzagliatin monotherapy. An empty square line represents C-peptide concentration, a filled triangle line represents GLP-1$_{total}$ concentration, and empty triangle line represents GLP-1$_{active}$ concentration. Data were presented as mean values ± SD. *$p < 0.05$, **$p < 0.01$ compared with combination treatment. iAUC$_{0-4h}$ incremental area under the curve for 4 h, OGTT oral glucose tolerance test.

**Table 3 | Pharmacodynamic parameters and surrogate estimates of glucose sensitivity during OGTT after sitagliptin, dorzagliatin, or sitagliptin + dorzagliatin treatment**

| | | Sitagliptin alone | Sitagliptin +Dorzagliatin | Dorzagliatin alone |
|---|---|---|---|---|
| PD parameters | | | | |
| Glucose | iC$_{max}$ (mg/dL) | 165.00 (34.70)* | 142.00 (44.90) | 163.00 (39.00) |
| | iC$_{av}$ (mg/dL) | 94.50 (22.00)** | 63.20 (29.00) | 84.80 (31.00)* |
| | iAUC$_{0-4h}$ (mg*h/dL) | 378.00 (87.80)** | 253.00 (116.00) | 339.00 (124.00)* |
| C-peptide | iC$_{max}$ (ng/mL) | 5.10 (2.40)* | 6.33 (3.77) | 4.15 (1.86)** |
| | iC$_{av}$ (ng/mL) | 3.04 (1.34) | 3.61 (1.80) | 2.46 (0.97)** |
| | iAUC$_{0-4h}$ (ng*h/mL) | 12.10 (5.35) | 14.40 (7.18) | 9.82 (3.86)** |
| GLP-1$_{total}$ | iC$_{max}$ (pmol/L) | 13.30 (18.50) | 9.17 (7.96) | 22.10 (13.20)** |
| | iC$_{av}$ (pmol/L) | 3.67 (3.89) | 2.81 (2.46) | 5.70 (2.94)* |
| | iAUC$_{0-4h}$ (pmol*h/L) | 14.70 (15.50) | 11.20 (9.85) | 22.80 (11.70)* |
| GLP-1$_{active}$ | iC$_{max}$ (pmol/L) | 8.24 (3.76) | 9.81 (6.17) | 6.64 (5.29) |
| | iC$_{av}$ (pmol/L) | 2.70 (1.48) | 3.06 (2.05) | 1.57 (0.93) |
| | iAUC$_{0-4h}$ (pmol*h/L) | 10.80 (5.93) | 12.20 (8.21) | 6.26 (3.73) |
| Glucose sensitivity | | | | |
| | $\Delta C_{30}/\Delta G_{30}$ (ng/mL per mg/dL) | 0.02*(0.01) | 0.04 (0.04) | 0.02*(0.01) |
| | $\Delta GLP\text{-}1_{total30}/\Delta G_{30}$ (pmol/L per mg/dL) | 0.12 (0.18) | 0.11 (0.10) | 0.23*(0.14) |
| | $\Delta GLP\text{-}1_{active30}/\Delta G_{30}$ (pmol/L per mg/dL) | 0.04 (0.008) | 0.09 (0.05) | 0.07 (0.05) |

Values are presented as the mean (SD).

$iC_{max}$ incremental maximum concentration from fasting state (at time of 0) before OGTT, $iC_{av}$ average concentration (calculated as iAUC$_{0-4h}$/4) from fasting state (at time of 0) before OGTT, $iAUC_{0-4h}$ incremental area under curve for 4 h, $OGTT$ oral glucose tolerance test, $\Delta C_{30}/\Delta G_{30}$ the iC at 30 min subtract 0 min of C-peptide divided by 30 min subtract 0 min glucose level, $\Delta GLP\text{-}1_{total30}/\Delta G_{30}$ the level at 30 min subtract 0 min of GLP-1$_{total}$ divided by 30 min subtract 0 min glucose level, $\Delta GLP\text{-}1_{active30}/\Delta G_{30}$, the level at 30 min subtract 0 min of GLP-1$_{active}$ divided by 30 min subtract 0 min glucose level.

*$p < 0.05$, **$p < 0.01$ compared with the combination treatment.

GLP-1 secretion in the C5757BL/6 J mice under OGTT conditions[29] and its signal on glucose-stimulated GLP-1 secretion in healthy Chinese subjects (ClinicalTrials.gov identifier: NCT01952535)[30]. Data from the whole-body radiography (WBR) in rats showed a high organ distribution of dorzagliatin in the pancreas, small intestine, and liver after an oral dose in the first 4 h, and the drug were then quickly cleared with a $t_{1/2}$ of 4.4 h in plasma. The mass balance study (ClinicalTrials.gov identifier: NCT03158506) was conducted in the US and results are consistent with fast clearance of dorzagliatin in humans and minimum renal clearance (less than 10%). We, therefore, suggest that the effect of dorzagliatin on glucose-stimulated GLP-1 secretion in patients with T2D and obesity is a combination of the role of GK in GLP-1 regulation and the pharmacokinetic property of dorzagliatin with intrinsic high organ distribution in the small intestine.

Unlike most of the clinical trials of dorzagliatin conducted in the population with T2D and non-obesity with BMI around 25 kg/m²[14–16], this study was conducted in patients with T2D and obesity in which dorzagliatin also showed its effectiveness in the glycemic control and improved insulinogenic effect and glucose sensitivity when combined with sitagliptin. There is no PK interaction between these two oral drugs and no increases in AEs. This offers an opportunity to treat patients with T2D and obesity through a combination of dorzagliatin and sitagliptin.

GK plays a central role in glucose homeostasis in pancreatic beta-cell, intestinal L cells, as well as hepatocytes[1]. The role of GK in the regulation of GLP-1 secretion has not been fully understood and the results from different studies are not consistent[31]. The most commonly accepted mechanism for GLP-1 secretion in L cells is supported by a sodium-glucose cotransporter-1 (SGLT-1) glucose transporter-regulated coupling of glucose-sodium flood into entero-L cells with the membrane depolarization through a K$_{ATP}$ channel blockage and voltage-gated calcium channel opening, which lead to GLP-1 release[32]. Theodorakis and colleagues reported that GK is expressed in human L and K cells, and may play an important role in incretin secretion in the entero-endocrine cells[33]. A significant increase of glucose-stimulated GLP-1 secretion by dorzagliatin alone suggested its effect on GK in entero-endocrine L cells. It is observed that the time for GLP-1 peak value of 30 min upon glucose load is correlated with the $T_{max}$ of dorzagliatin exposure (Supplementary Fig. S4).

Ferrannini and colleagues reported that GLP-1 secretion quickly reached the maximum level of GLP-1$_{total}$ in healthy Caucasians in a range of 25–30 pmol/L within 30 min after 75-g glucose oral mixed meal, compared with a $C_{max}$ of 10–12 pmol/L for T2D subjects with obesity[11], although it was reported that the GLP-1 and GIP secretion in the GCK-MODY patients was not impaired in these subjects with non-obesity[31,34]. The inconsistency in the GLP-1 secretion in GCK-MODY and

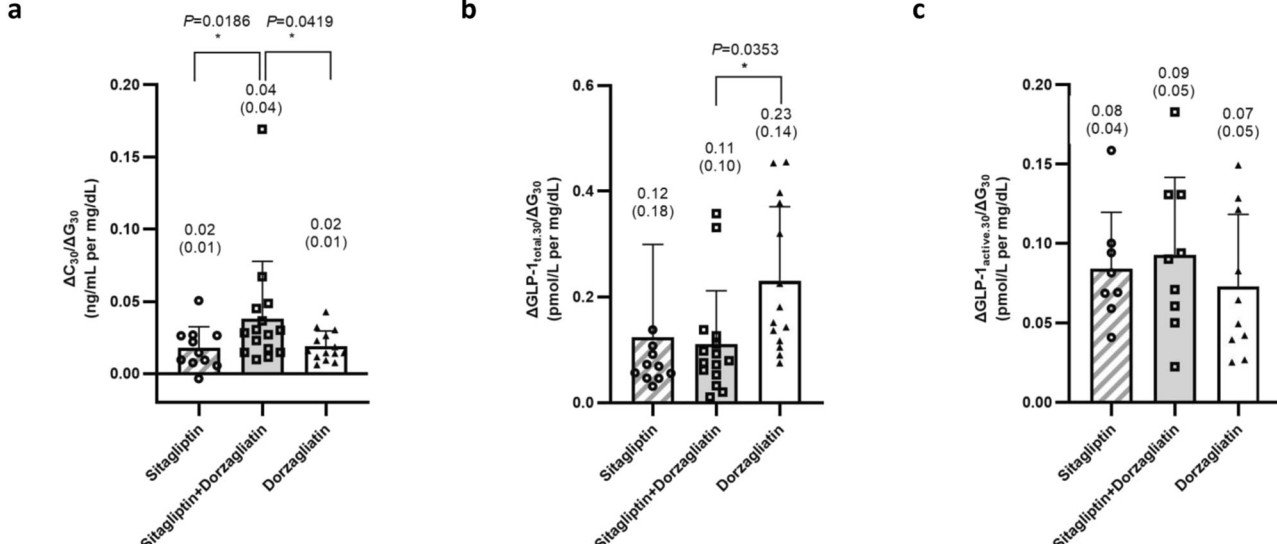

Fig. 2 | **Glucose sensitivity. a** The insulinogenic index ($\Delta C_{30}/\Delta G_{30}$) (sitagliptin, $n = 11$; sitagliptin + dorzagliatin, $n = 15$; dorzagliatin, $n = 14$), **b** total GLP-1 secretion index ($\Delta GLP\text{-}1_{total.30}/\Delta G_{30}$) (sitagliptin, $n = 11$; sitagliptin + dorzagliatin, $n = 15$; dorzagliatin, $n = 14$), and **c** active GLP-1 secretion index ($\Delta GLP\text{-}1_{active.30}/\Delta G_{30}$) (sitagliptin, $n = 8$; sitagliptin + dorzagliatin, $n = 9$; dorzagliatin, $n = 10$) are calculated based on the C-peptide and GLP-1 levels in the OGTT study. A diagonal bar and empty circles represent the mean and individual index under sitagliptin

monotherapy, gray bar, and empty squares represent the mean and individual index under sitagliptin + dorzagliatin therapy, white bar and empty triangles represent the mean and individual index under dorzagliatin monotherapy. Error bars represent standard deviation. The *p* value was calculated based on the paired Wilcoxon test between combination therapy and each monotherapy. \**p* < 0.05, \*\**p* < 0.01 compared with combination treatment. OGTT oral glucose tolerance test.

T2D could arise from the expression state of GK in T2D vs GCK-MODY subjects. A substantial reduction of GK expression in the pancreas and liver in T2D patients has been reported, which leads to the loss of glucose sensitivity in the pancreas and reduced hepatic glycogen production[24,25,35]. In the current study, dorzagliatin alone improved the impaired glucose-stimulated GLP-1 secretion in patients with T2D and obesity with an iC$_{max}$ of GLP-1$_{total}$ reached 22.10 pmol/L (-27 pmmol/L in C$_{max}$, Supplementary Fig. S4), approaching to the healthy range of maximum GLP-1 level described above.

Recently, GLP-1 and its most abundant inactive GLP-1 metabolite GLP-1 (9-36)NH2 has been reported to have biological activities in protecting human aortic endothelial cells and cardiomyocytes in vivo and ex vivo studies[36–38]. The cleaved peptide is found in almost twofold magnitude higher concentrations than active GLP-1 in peripheral blood and shows cardioprotection, and antioxidant properties[31] as well as demonstrates the capability to inhibit hepatic neoglucogenesis[39]. The benefits of GLP-1 and metabolites regulated by dorzagliatin shall be further evaluated when it is used for the treatment of T2D as a monotherapy.

The fast clearance of GLP-1 and relatively low concentration of GLP-1$_{active}$ in the circulation can be explained by the effect of DPP-4 activity. However, the pharmacological effect of dorzagliatin on GLP-1 secretion measured by GLP-1$_{total}$ was substantially suppressed by the addition of sitagliptin. The increase of GLP-1$_{active}$ by the combination of dorzagliatin and sitagliptin compared with either monotherapy should result from the DPP-4 inhibition effect of sitagliptin to slow down the GLP-1 degradation under an increased GLP-1 production modulated by dorzagliatin. It has been reported that elevated GLP-1$_{active}$ concentrations restrict the GLP-1 secretion to some degree[40–43]. This feedback effect has also been observed in other studies in humans, where an exogenous infusion of active GLP-1 (7–37) led to a reduction in levels of endogenous GLP-1[44], and in the clinical trial studying sitagliptin[45]. Brubaker and Hansen have reported that GLP-1 can stimulate somatostatin release from isolated rat intestinal cultures[46] and somatostatin inhibits GLP-1 secretion, indicating that GLP-1 limits its own secretion through a somatostatin-mediated paracrine-inhibitory pathway[47].

Although the GLP-1$_{total}$ level is reduced in the combination treatment with dorzagliatin and sitagliptin, the overall secretion of C-peptide in response to glucose challenge has been significantly improved over monotherapies in this study. This effect may result from the GLP-1 secretion from pancreatic alpha-cell through a paracrine mode of action on beta-cell.

In conclusion, dorzagliatin regulates glucose homeostasis not only via its dual-activating GK activities in the pancreas and liver but also through the improvement of glucose-stimulated GLP-1 release in T2D patients. It increases the effectiveness of glycemic control and glucose sensitivity when combined with a DPP-4 inhibitor sitagliptin. The lack of PK interaction between dorzagliatin and sitagliptin further supports the combination of dorzagliatin with sitagliptin for the treatment of patients with T2D and obesity.

## Methods

This study was designed to evaluate the PK and PD effects of dorzagliatin and sitagliptin either alone or in combination in patients with T2D and obesity who were on standard anti-diabetes drug (clinical trials identifier: NCT03790839). The protocol was approved by an Institutional Review Board (IntegReview Ethics Review Board, Austin, USA) at the study site, and conducted in accordance with the Declaration of Helsinki and International Conference on Harmonization Good Clinical Practice (ICH-GCP) guidelines, as well as US Food and Drug Administration regulations. All participants provided written informed consent prior to participating in the study and were compensated for completed study procedures. The patient enrollment was from 21 December 2018 to 30 August 2019.

### Key eligibility criteria/study population

Male and female adults eligible for inclusion had to meet the following criteria: patients aged between 30 and 65 years, in general, good health who had been diagnosed as T2D for at least 3 months, with HbA1c between 7.0 and 10.5%, body mass index (BMI) between 19.0 and 38.0 kg/m$^2$, taking a stable dose of metformin ≥1000 mg per day, or a DPP-4 inhibitor, or a sodium-glucose cotransporter-2 (SGLT-2)

inhibitor, or metformin plus a DPP-4 inhibitor with no change in the dose for at least 4 weeks prior to screening, and accepting to change their current therapy to 100 mg sitagliptin QD for at least 14 days prior to dosing on Day 1.

The key exclusion criteria included: fasting blood glucose (FBG) ≤110 or ≥270 mg/dL, the reported incidence of severe or serious hypoglycemia within 3 months prior to screening, type 1 diabetes or latent autoimmune diabetes, known hypersensitivity/contraindication to study drugs, evidence of any clinically significant medical illness or functional disorders, and pregnant or breast-feeding women.

## Study design

This is a phase 1, open-label, single-sequence, multiple-dose study conducted at a single clinical center in the US (Frontage Clinical Services, Inc., Secaucus, NJ).

Eligible subjects had a minimum 12-day sitagliptin run-in period (sitagliptin 100 mg QD) prior to admission to the clinical research center, and each subject completed the medical diary to record study drug (sitagliptin) doses taken every day and the results of the blood glucose monitoring. Following completion of the run-in period, eligible subjects were admitted to the clinical research center on Day −2 for a total of 18 overnight stays, and discharged after completion of end-of-study (EOS) procedures on Day 17. All eligible subjects received 3 treatment regimens sequentially as shown in Supplementary Fig. S1: sitagliptin 100 mg QD for 5 days (Day 1–5), then sitagliptin 100 mg QD + dorzagliatin 75 mg BID for 5 days (Day 6–10), followed by dorzagliatin 75 mg BID for 5 days (Day 11–15). Only the morning dose was administered, and sampling was performed on Days 5, 10, and 15 for up to 24 h post-dose. All treatments were administered 60 min prior to meals except on Days 5, 10, and 15; when OGTT was conducted, subjects rapidly (within 5 min) drank a solution containing 75-g glucose 30 min after study drug oral administration instead of a breakfast.

The primary outcomes of this study include potential PK drug–durg interaction between dorzagliatin and sitagliptin by GMR, and the assessment of safety and tolerability of dorzagliatin with simultaneous administration of sitagliptin in subjects with T2D. The secondary outcomes include the PD responses of glucose, GLP-1, and C-peptide by $iAUC_{0-4h}$ and $C_{max}$, and glucose sensitivity following dorzagliatin, sitagliptin, or simultaneous administration of dorzagliatin and sitagliptin in subjects with T2D.

Sample size calculations were based on study design and intra-subject variability. At least ten evaluable subjects in the sequence would be required to achieve a power of at least 0.8 for GMR between two treatments (sitagliptin + dorzagliatin vs. dorzagliatin alone or sitagliptin + dorzagliatin vs. sitagliptin alone) for $C_{max}$ or $AUC_{0-24h}$, with the equivalence bounds of 0.8 to 1.25. Assuming a drop-out rate of 20%. 15 eligible subjects were planned to enroll by aiming to obtain 12 evaluable subjects for pharmacokinetic drug–drug interaction assessment.

## PK and PD sample collection

Blood samples for PK analysis were collected in dipotassium ethylenediaminetetra acetic acid ($K_2EDTA$) tubes on Days 5, 10, and 15 at predose, 0.25, 0.5, 1, 1.5, 2, 3, 4, 6, 8, 10, 12, 18, and 24 h post-dose.

In the OGTT studies on Days 5, 10, and 15, 75 g of glucose was administered orally 30 min after experimental drug administration, and the blood samples were collected at pre-dose, 0.5, 1, 1.5, 2, 2.5, 3, and 4 h post-oral glucose intake. The samples were used to measure serum glucose, C-peptide, and plasma GLP-1. The preparation of the sample is detailed in the supplementary material.

## Analytical methods

The concentrations of dorzagliatin and sitagliptin in plasma were determined by using liquid chromatography-tandem mass spectrometry (LC-MS/MS) methods (Frontage Laboratories, Inc., Pennsylvania, USA) (refer to supplementary material for details).

Serum glucose was analyzed using a hexokinase enzymatic method, serum C-peptide was assessed using a chemiluminescence assay (BioReference Laboratories, Inc., New Jersey, USA). The determinations of plasma GLP-1 concentrations were performed by using validated enzyme-linked immunosorbent assays (ELISA) (Mercodia AB, Uppsala, Sweden) (refer to supplementary material for details).

## PK and PD assessment

The non-compartmental analysis was applied to determine the PK using WinNonlin software (Certara, Princeton, NJ, USA). PK parameters were derived from the plasma concentration-time curve, $C_{max}$ and $T_{max}$ were directly determined from the plasma concentration-time profile of each subject. The $AUC_{0-24h}$ was calculated using the linear trapezoidal method.

PD parameters were evaluated by measurement of serum glucose, C-peptide, and plasma GLP-1 concentrations, using the incremental area under curve for 4 h, $iAUC_{0-4h}$, incremental maximum concentration, $iC_{max}$, and average concentration ($iC_{av}$, calculated as $iAUC_{0-4h}/4$) from fasting state (at time of 0) before OGTT.

## Safety assessment

Safety evaluations were conducted throughout each study based on clinical laboratory tests, vital signs, physical examinations, 12-lead ECG, and AE. Any AE reported, especially TEAE, was recorded and coded using the Medical Dictionary for Drug Regulatory Activities (MedDRA), and its relationship to the drug treatment was determined by the investigator.

All AEs were monitored after the administration of the study drugs.

## Statistical analysis

For statistical analysis, subjects who received at least one dose of the study drug and had at least one post-enrollment safety assessment were included in the safety analysis set. The PK analysis set included the subjects who had no major protocol deviations and had sufficient PK data to obtain estimates of key PK parameters. The PD analysis set included those subjects who had sufficient glucose, C-peptide, and GLP-1 concentrations to obtain estimates of PD parameters.

Mixed models under the sequential design were used to analyze $C_{max}$ and $AUC_{0-24h}$ with treatments as fixed effects and subjects as random effects. The GMR (combination/monotherapy ratio) and the corresponding 90% CI for $C_{max}$ and $AUC_{0-24h}$ were obtained by exponentiating the adjusted mean difference in logarithms. The incremental PD parameters were calculated by subtracting the fasting state (time of 0 before OGTT) value from the values at each sampling time-point. A comparison was performed using two mixed models corresponding to sitagliptin + dorzagliatin versus sitagliptin, or sitagliptin + dorzagliatin versus dorzagliatin. P values were calculated based on the test of the ratio (log-difference) from the models.

Parameters for glucose sensitivity are evaluated through the C-peptide index ($\Delta C_{30}/\Delta G_{30}$) and GLP-1 index ($\Delta GLP\text{-}1_{total.30}/\Delta G_{30}$, $\Delta GLP\text{-}1_{active.30}/\Delta G_{30}$), which were accessed through the level at 30 min subtract 0 min of C-peptide, GLP-1$_{total}$, and GLP-1$_{active}$ respectively and then divide by 30 min subtract 0 min glucose level, which data for calculating were all collected in OGTT. The $p$ value was calculated based on the paired Wilcoxon test between combination therapy and each monotherapy.

## Reporting summary

Further information on research design is available in the Nature Portfolio Reporting Summary linked to this article.

## Data availability

The data from this study cannot be made publicly available due to the sponsor's contractual obligations. We encourage researchers or parties interested in collaboration for non-commercial use to submit an application to the corresponding author (lichen@huamedicine.com). Applications should specifically outline the data the parties are interested in receiving and how the data will be used; the use of the data must also comply with the country- or region-specific regulations. A signed data access agreement with the sponsor is required before accessing the shared data. Individual participant data under the results reported in this article can be de-identified and provided by this mechanism. All shared data will be available beginning immediately and ending 24 months following the publication of this article. The study protocol is included in Supplementary Note 1. Source Data for the data presented in graphs within the figures in this manuscript are provided with this paper. Source data are provided with this paper.

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

## Acknowledgements

The authors thank all the patients enrolled in this study, site investigators, and coordinators who participated and contributed to this study. The study sponsor was Hua Medicine. We thank C. Chen (Hua Medicine) for assistance with the data analysis; A. Wang (Hua Medicine) for assistance with the safety data; and G. Yu and F. Tang, who reviewed an earlier version of the manuscript on behalf of Hua Medicine. Hua Medicine participated in the design, conduct, data analysis and interpretation of the clinical study, the preparation of the manuscript, and the decision to publish.

## Author contributions

L.C. contributed to the study design, G.J.T. and J.Z. contributed to patient recruitment and study conduction, Y.S., Q.Z., and J.Z. contributed to drafting the manuscript. X.L., Z.F., and L.F. contributed to the data analysis. Y.Z. and B.H. contributed to medical monitoring and pharmacovigilance in the study. L.C., J.Z., and Y.S. contributed to the interpretation of the data. All authors approved the final version for publication.

## Competing interests

The authors L.C., J.Z., Y.S., Y.Z., X.L., Z.F., L.F., B.H., and Q.Z. are employees of Hua Medicine. G.J.T. is the leading investigator who conducted the study, declaring no competing interests.
