## [Peer Review File · Nature Communications]

A phase I open-label clinical trial to study drug-drug interactions of Dorzagliatin and Sitagliptin in patients with type 2 diabetes and obesityREVIEWER COMMENTS

Reviewer #1 (Remarks to the Author):

In this rather small study of a very heterogeneous disease (type 2 diabetes), the combination of sitagliptin and dorzagliatin is found to be safe and reduce plasma glucose excursions following an OGTT as well as increase C-peptid/insulin secretion and glucose sensitivity of the beta cells. The work contributes with the information of successful combination of two anti-diabetes compounds but does not discuss the potential mechanism or expected side effects if used long-term or in a larger population.

The methodologies used are well established and acknowledged.

Have you performed power calculations? What was the primary outcome?

The language needs a bit more work, the abbreviations are not all explained, and especially figures 1e, 1f and 1g are difficult to interpret (no labels explain the difference between the three figures).

I really miss an exploration of the mechanisms behind the higher GLP-1 levels after dorzagliatin treatment and also a more thorough introduction to the two drugs, the rationale of the study and perspectives on this combined treatment. 13 lines of introduction is too brief.

Reviewer #2 (Remarks to the Author):

Dorzagliatin is an orally active glucokinase activator (GKA) which acts on glucokinase (GK) in the pancreas and liver for the treatment of T2D (Refs. 12-15). In the present study, the authors reported its role in the regulation of GLP-1 release in response to oral glucose challenge in obese people with type 2 diabetes either alone or in combination with sitagliptin, a DDP-4 inhibitor, and proposed a triple acting role of dorzagliatin in regulation of glucose homeostasis.

Criticisms

1. Importantly, the combination treatment of dorzagliatin with sitagliptin achieved a greater glucose lowering effect than sitagliptin or dorzagliatin monotherapy. The authors focused on GLP-1 and pancreatic beta cells, but it is unclear to this reviewer how the role of GK in hepatocytes is involved in this process.
2. A significant increase of iC_{max} of glucose-stimulated GLP-1 secretion (GLP-1_{total}) was observed in dorzagliatin monotherapy. However, such increase was not observed in dorzagliatin plus sitagliptin therapy (Figure S2). How do the authors explain the discrepancy?
3. The role of GK in the regulation of GLP-1 secretion has not been fully understood and the results from different studies are not consistent (Ref. 19). In the present study, the authors clearly demonstrated a significant increase of glucose-stimulated GLP-1 secretion by dorzagliatin alone. However, it seems unclear to this reviewer whether this effect is common with GKAs or specific to dorzagliatin. Information on the impact of other GKAs on GLP-1 secretion is highly appreciated.
4. Dorzagliatin regulates glucose-stimulated GLP-1 release and synergizes with sitagliptin in optimizing glycemic control in obese people with type 2 diabetes. Is this effect seen in lean people with type 2 diabetes? Comment or preliminary data is highly appreciated.

Reviewer #3 (Remarks to the Author):

Summary: While this was an open-label, single-arm study of N=15 subjects, it was a carefully designed and well-executed experiment. The lab measurements taken were informative and the analyses appropriate. The writing was clear and the supplementary material helpful. The data support the primary claim that dorzagliatin works well in combination with sitagliptin to improve glycemic control as measured in this study.

Moderate concern with the sample size: The small sample size is understandable given the intensity of the data collection employed. However, the small sample size did limit the study's precision and strength of the evidence. Only large effects were able to reach statistical significance and some of the significant comparisons only reached a 0.05 threshold. 90% confidence intervals were presented, possibly as an acknowledgement of the limited precision of the estimates. I personally found the study compelling, but I was not a skeptic of the main arguments being tested. A more skeptical reader might not be swayed by well-done but small study like this one.

Minor concern with the term synergistic: The title claims "Dorzagliatin ... is synergistic with sitagliptin in glycemic control". The term synergistic implies that the use of the two medications together produces an effect that is greater than the sum of the individual effects. I did not see an analysis supporting this claim. Indeed, the study design is unable to test this claim. That would require glycemic measurements with the subjects on neither drug and a test of whether the effect of the two medications together was multiplicative or just additive. Without this comparison, it cannot be ruled out that the two medications together are equal to or even less than the sum of the individual effects. The study did show the combination of medications was more effective than either medication alone and showed distinct mechanisms of action. The term synergistic seems to apply colloquially, but it was unclear if it applies scientifically.

Minor concern with lines 146-47: Please clarify what is meant by "which was either statistically or numerically increased in combination treatment".

Minor concern with the study design section: Please clarify if the subjects stayed on their other anti-diabetic medications (metformin, DPP4, etc.) during the experimental phase. I assumed yes and that is certainly implied by the eligibility criteria, but I didn't see it explicitly stated in the study design.

Answers to the questions from reviewers

Li Chen

20221115

REVIEWER COMMENTS

Reviewer #1 (Remarks to the Author):

In this rather small study of a very heterogenous disease (type 2 diabetes), the combination of sitagliptin and dorzagliatin is found to be safe and reduce plasma glucose excursions following an OGTT as well as increase C-peptid/insulin secretion and glucose sensitivity of the beta cells.

The work contributes with the information of successful combination of two anti-diabetes compounds but does not discuss the potential mechanism or expected side effects if used long-term or in a larger population.

Reply to the comments: we appreciate your comments and your questions on the mechanism or expected side effects if used for a longtime. Dorzagliatin has been studied in two phase III trials in drug naïve and metformin tolerated T2D patients for a 52-week observation on its safety and tolerability. It is well tolerated with low hypoglycemia rate less than 1% and minimum safety signals were detected during these studies as reported in the Nature Medicine this May [D Zhu ... L Chen Nature Med 2022; 28:965; W Yang ... L Chen Nature Medicine 2022; 28:974; John Buse Nature Medicine 2022; 28:901]. Recent work from Julianna Chan and Elaine Chow on the genetically proxied GK activation suggested that GKAs may protect against coronary artery disease (CAD) and Heart Failure (HF) [Cardiovascular Diabetology 2022; 21:192] and post marketing real world study on its long-term benefit and risk will be performed. The combination of dorzagliatin and sitagliptin will be conducted after dorzagliatin launch to investigate the immunological effect of DPPIV inhibitor in Chinese T2D patients, otherwise we don't anticipate new side effects that the combination therapy will encounter. We include a paragraph discussing the mechanism of action, safety and tolerability of dorzagliatin and sitagliptin in the introduction in line 54-79 of our article.

The methodologies used are well established and acknowledged.

Have you performed power calculations? What was the primary outcome?

Reply to the Question: The primary objectives of this study are to assess the potential PK interaction between dorzagliatin and sitagliptin, and to evaluate the safety and tolerability of dorzagliatin with simultaneous administration of sitagliptin in subjects with T2DM. The secondary objective of this study is to assess the pharmacodynamic (PD) responses of PD markers, such as glucose, GLP-1, and C-peptide, following dorzagliatin, sitagliptin, or simultaneous administration of dorzagliatin and sitagliptin in subjects with T2DM.

Sample size calculations were based on study design and intra-subject variability. At least 10 evaluable subjects in the sequence would be required to achieve a power of at least 0.8 for the geometric mean ratios (GMR) between two treatments (sitagliptin+dorzagliatin vs. dorzagliatin alone or sitagliptin+dorzagliatin vs. sitagliptin alone) for C_{max} or AUC_{0-24h} , with

the equivalence bounds of 0.8 to 1.25. Assuming a drop-out rate of 20%, we planned to enroll 15 eligible subjects by aiming to obtain 12 evaluable subjects for pharmacokinetic Drug-Drug Interaction assessment.

For the power of glucose difference between sitagliptin and sitagliptin+dorzagliatin treatment groups, under the sample size based on the primary endpoint $n = 15$; the difference of glucose $iAUC_{0-4h}$ between sitagliptin and sitagliptin+dorzagliatin treatment is $378-253 = 125 \text{ mg}^*h/dL$, standard deviation of sitagliptin and sitagliptin+dorzagliatin treatment is 87.8 and 116 mg^*h/dL respectively; in two-sided test at a significant level of 0.01, the power is 0.94. The calculation of effect size was based on Cohen's d .

For the power of total GLP-1 difference between dorzagliatin and sitagliptin+dorzagliatin treatment groups, sample size $n = 15$; the difference of total GLP-1 $iAUC_{0-4h}$ between dorzagliatin and sitagliptin+dorzagliatin treatment is $22.8-11.2 = 11.6 \text{ pmol}^*h/L$, standard deviation of sitagliptin and sitagliptin+dorzagliatin treatment is 11.7 and 9.85 pmol^*h/L respectively; in two-sided test at a significant level of 0.05, the power is 0.97. The calculation of effect size was based on Cohen's d .

The language needs a bit more work, the abbreviations are not all explained, and especially figures 1e, 1f and 1g are difficult to interpret (no labels explain the difference between the three figures).

Reply to the comments: Thank you for your suggestion on the language and abbreviations. We have now corrected them throughout the manuscript. The legends to figures were revised to explain the figures (legends to figures is in line 474-490).

I really miss an exploration of the mechanisms behind the higher GLP-1 levels after dorzagliatin treatment and also a more thorough introduction to the two drugs, the rationale of the study and perspectives on this combined treatment. 13 lines of introduction is too brief.

Reply to the comments: Yes, we agree with your suggestion and rewrite the introduction session in line 38-79. We discussed the mechanisms behind the higher GLP-1 levels after dorzagliatin treatment in line 208-217

Reviewer #2 (Remarks to the Author):

Dorzagliatin is an orally active glucokinase activator (GKA) which acts on glucokinase (GK) in the pancreas and liver for the treatment of T2D (Refs. 12-15). In the present study, the authors reported its role in the regulation of GLP-1 release in response to oral glucose challenge in obese people with type 2 diabetes either alone or in combination with sitagliptin, a DDP-4 inhibitor, and proposed a triple acting role of dorzagliatin in regulation of glucose homeostasis.

Criticisms

1. Importantly, the combination treatment of dorzagliatin with sitagliptin achieved a greater glucose lowering effect than sitagliptin or dorzagliatin monotherapy. The authors focused on GLP-1 and pancreatic beta cells, but it is unclear to this reviewer how the role of GK in hepatocytes is involved in this process.

Reply to the comments: Thank you for raising an important question on how hepatic GK is involved in the glucose homeostasis together with its action on glucose stimulated insulin and GLP-1 secretion. We discussed this in line 38-49 in the introduction session as the following. "Improvement of hepatic glucose metabolism by dorzagliatin is found in relationship with effective postprandial glucose reduction and will be further studied in subsequent trials [R Basu, ClinicalTrials.gov Identifier: NCT05098470]. It has been reported that substantial reduction of hepatic GK expression in T2D patients correlated with their hyperglycemia and hepatic insulin resistance [A Basu, Diabetes 2001; 50:1351; R Haeusler Mol Metabolism 2015; 4:222]. Epigenetic modification of hepatic GCK promoter caused a reduction of hepatic GCK expressions and reduced glycogen in the liver [M Jiang, Diabetology, 2008; 51(8):1525-1533]. Dorzagliatin restored the hepatic GK expression in diabetes rats and improved glycemic control after 28-day treatment [P Wang J Diabetes Res, 2017, AID 5812607]." The discussion is added in line 183-202, and line 220-225.

2. A significant increase of iCmax of glucose-stimulated GLP-1 secretion (GLP-1total) was observed in dorzagliatin monotherapy. However, such increase was not observed in dorzagliatin plus sitagliptin therapy (Figure S2). How do the authors explain the discrepancy?

Reply to the comments: Thank you for pointing out this very important finding in this study. For the short answer, sitagliptin is known to inhibit the total GLP-1 secretion through an active GLP-1 mediated self-control feedback loop. That is when DPP-4 inhibitor increases the active GLP-1 level, the active GLP-1 will inhibit the glucose stimulated GLP-1 secretion. The lengthier explanation can be seen in the discussion session (in line 239-244) in this manuscript. Earlier research by Deacon showed that DPP-4 inhibitor preserved the active form of GLP-1 but suppressed meal-induced incretin secretion in dogs [Deacon, J Endocrinol 2002; 172:355]. Infusion of exogenous active GLP-1 also reduced levels of endogenous GLP-1, suggesting the feed-back control [Toft-Nielsen, Diabetes Care 1999; 22:1137].

3. The role of GK in the regulation of GLP-1 secretion has not been fully understood and the results from different studies are not consistent (Ref. 19). In the present study, the authors clearly demonstrated a significant increase of glucose-stimulated GLP-1 secretion by dorzagliatin alone. However, it seems unclear to this reviewer whether this effect is common with GKAs or specific to dorzagliatin. Information on the impact of other GKAs on GLP-1 secretion is highly appreciated.

Reply to the comments: It is indeed a very important question, and we believe the effect is unique to dorzagliatin. In the preclinical studies we discovered its effect on glucose stimulated GLP-1 secretion in the C5757BL/6J mice under OGTT conditions [ADA 2014, 134-LB] and its signal on glucose stimulated GLP-1 secretion in healthy Chinese subject [ClinicalTrials.gov Identifier: NCT01952535, ADA 2015, 1165-p]. Data from the Whole-Body Radiography (WBR) in rats showed a high organ distribution of dorzagliatin in the pancreas, small intestine and liver after oral dose in the first 4 hours and the drug were then quickly cleared with a t_{1/2} of 4.4 hours in plasma. The mass balance study [ClinicalTrials.gov Identifier: NCT03158506] was conducted in US and results are consistent with fast clearance of dorzagliatin in human and renal clearance is less than 10%. We therefore suggest that the effect of dorzagliatin on glucose stimulated GLP-1 secretion in obese T2D subjects is a combination of the role of GK in GLP-1 regulation and the pharmacokinetic property of dorzagliatin. Based on your recommendations, we have searched literature and have not seen reports of other GKA's effect on GLP-1 secretion.

4. Dorzagliatin regulates glucose-stimulated GLP-1 release and synergizes with sitagliptin in optimizing glycemic control in obese people with type 2 diabetes. Is this effect seen in lean people with type 2 diabetes? Comment or preliminary data is highly appreciated.

Reply to the comments: This is an excellent question. Based on our understanding, glucose stimulated GLP-1 secretion in the obese IGT and T2D subjects is reduced dramatically compared with NGT [Muscelli, Holst and Ferrannini et al Diabetes 2008; 57, 1340], although Dr Knop reported a lack of difference in the IGT and T2D subjects from NGT in general [Knop Diabetologia 2013;56:965]. Larsen and Torekov conducted a review lately on 19 publications in which they concluded that several factors, such as BMI, Age and Metformin use, were affecting the GLP-1 secretion [Larsen and Torekov, J Diabetes Res 2017; AID 7583506]. The state of obesity is an independent factor from IGT and T2D that impacts on the glucose stimulated GLP-1 secretion. Singh reviewed GLP-1 secretion status in healthy and T2D subjects in Asian compared with Caucasian recently and discussed the ethnic difference in the fasting and glucose stimulated GLP-1 secretion [Singh, Indian Journal of Endocrinol and Metabol 2015; 19: 30-38]. Dysfunction in glucose stimulated GLP-1 secretion function was associated with East Asian but to a less degree in South Asian in which insulin resistance is the major cause of T2DM. In our single ascending dose study of dorzagliatin in Chinese healthy volunteers [ClinicalTrials.gov Identifier: NCT01952535] with BMI of 21.6 kg/m², we evaluated the effect of dorzagliatin on glucose stimulated GLP-1 secretion and found an increase of total GLP-1 secretion at high dose group of subjects (25, 35 and 50 mg) over placebo [ADA 2014, 134-LB]. This prompted us to revisit GLP-1 topic in the combination study with DPPIV inhibitor reported here [ClinicalTrials.gov Identifier: NCT03790839] and confirms our earlier observation in China NGT subjects.

Reviewer #3 (Remarks to the Author):

Summary: While this was an open-label, single-arm study of N=15 subjects, it was a carefully designed and well-executed experiment. The lab measurements taken were informative and the analyses appropriate. The writing was clear and the supplementary material helpful. The data support the primary claim that dorzagliatin works well in combination with sitagliptin to improve glycemic control as measured in this study.

Moderate concern with the sample size: The small sample size is understandable given the intensity of the data collection employed. However, the small sample size did limit the study's precision and strength of the evidence. Only large effects were able to reach statistical significance and some of the significant comparisons only reached a 0.05 threshold. 90% confidence intervals were presented, possibly as an acknowledgement of the limited precision of the estimates. I personally found the study compelling, but I was not a skeptical of the main arguments being tested. A more skeptical reader might not be swayed by well-done but small study like this one.

Reply to the comments: Thank you for your comment. Sample size was calculated based on study design and intra-subject variability for the determination of pharmacokinetic interaction between sitagliptin and dorzagliatin. At least 10 evaluable subjects in the sequence would be required to achieve a power of at least 0.8 for the geometric mean ratios between two treatments (sitagliptin+dorzagliatin vs. dorzagliatin alone or sitagliptin+dorzagliatin vs. sitagliptin alone) for C_{max} or AUC_{0-24h} of the testing drugs, with the equivalence bounds of 0.8 and 1.25. Assuming a drop-out rate of 20%, we would plan to enroll 15 eligible subjects by aiming to obtain 12 evaluable subjects for pharmacokinetic drug-drug interaction assessment.

We fully understand your concern and have calculated the power of PD markers. The power of glucose difference between sitagliptin and sitagliptin+dorzagliatin treatment groups is 0.94. The power of total GLP-1 difference between dorzagliatin and sitagliptin+dorzagliatin treatment groups is 0.97.

Minor concern with the term synergistic: The title claims "Dorzagliatin ... is synergistic with sitagliptin in glycemic control". The term synergistic implies that the use of the two medications together produces an effect that is greater than the sum of the individual effects. I did not see an analysis supporting this claim. Indeed, the study design is unable to test this claim. That would require glycemic measurements with the subjects on neither drug and a test of whether the effect of the two medications together was multiplicative or just additive. Without this comparison, it cannot be ruled out that the two medications together are equal to or even less than the sum of the individual effects. The study did show the combination of medications was more effective than either medication alone and showed distinct mechanisms of action. The term synergistic seems to apply colloquially, but it was unclear if it applies scientifically.

Reply to the comments: We highly appreciate your comments on the term of synergistic. Agree with you that we can not provide data that support the synergy claim and would like to change it into "better glycemic control combined with sitagliptin in obese T2D" in the title in line 3-4

Actually, the synergistic effect in this article is more inclined to explain in a general sense that the two drugs are complementary and cooperative from different mechanisms, but not restricted in glucose reduction.

This synergistic effect (complementary and cooperative) in mechanism contributed to the improvement of the early-phase C-peptide secretion and GLP-1 secretion, indicating enhanced beta-cell function (refer to the fig below, which is also illustrated in Figure 2 and Table 2 in the manuscript).

Minor concern with lines 146-47: Please clarify what is meant by “which was either statistically or numerically increased in combination treatment”.

Reply to the comments: Thank you very much for your comments. We have made the following changes in the manuscript for better clarity. “Accordingly, the enhanced beta-cell secretion function was associated with an improved GLP-1 secretion index in combination treatment (Figure 2 and Table 2).” in line 176-177.

Minor concern with the study design section: Please clarify if the subjects stayed on their other anti-diabetic medications (metformin, DPP4, etc.) during the experimental phase. I assumed yes and that is certainly implied by the eligibility criteria, but I didn't see it explicitly stated in the study design.

Reply to the comments: Thank you for the comments. We shall have given a better description in the eligibility criteria “... .. accepting to change their current therapy to 100 mg sitagliptin QD for at least 14 days prior to dosing on Day 1.” in line 268-269. The protocol defined the eligible participants were those already on metformin, DPP4 inhibitors, SGLT2 inhibitors or combination of metformin and either DPP4 or SGLT2 antidiabetics. The inclusion criteria#2 specified the eligible subjects must change their antidiabetic therapy to 100 mg sitagliptin QD for at least 14 days prior to dosing on Day 1. Once the patients were eligible, they were enrolled into a minimum 12-day run-in period and switched antidiabetic medications to sitagliptin 100 mg QD regimen to standardize the control medication co-administered with dorzagliatin.

REVIEWER COMMENTS

Reviewer #1 (Remarks to the Author):

Thank you for the sufficient replies to my questions.

I would recommend to include the results of the sample size calculations and informations about primary outcome in the manuscript - or supplementary material.

Moreover, I believe that it is timely to change the wording throughout the manuscript incl. the title: 'Patients with type 2 diabetes and obesity' instead of 'obese patients with type 2 diabetes' and so forth.

Reviewer #2 (Remarks to the Author):

Dorzagliatin is an orally active glucokinase activator (GKA) which acts on glucokinase (GK) in the pancreas and liver for the treatment of type 2 diabetes. In the present study, the authors reported its role in the regulation of GLP-1 release in response to oral glucose challenge in obese people with type 2 diabetes either alone or in combination with sitagliptin, a DDP-4 inhibitor, and proposed a triple acting role of dorzagliatin in regulation of glucose homeostasis. I have no further comments.

REVIEWERS' COMMENTS

Reviewer #1 (Remarks to the Author):

Thank you for the sufficient replies to my questions.

I would recommend to include the results of the sample size calculations and informations about primary outcome in the manuscript - or supplementary material.

Moreover, I believe that it is timely to change the wording throughout the manuscript incl. the title:

'Patients with type 2 diabetes and obesity' instead of 'obese patients with type 2 diabetes' and so forth.

Response to Reviewer:

Thank you for the comments, we totally agree with your suggestion.

A detail description of sample size calculation and primary outcomes is edited in the Supplementary information- "Supplementary methods"- "Sample size calculation", "Outcomes".

The title is revised as "A phase I open-label clinical trial to study drug-drug interactions of Dorzagliatin and Sitagliptin in patients with type 2 diabetes and obesity" at line 3-4, and changed into "patients with T2D and obesity" throughout the manuscript at line of 24, 26, 36-37, 79, 201-202, 204, 205, 208, 222, 223, 228, 256, 260.

Reviewer #2 (Remarks to the Author):

Dorzagliatin is an orally active glucokinase activator (GKA) which acts on glucokinase (GK) in the pancreas and liver for the treatment of type 2 diabetes. In the present study, the authors reported its role in the regulation of GLP-1 release in response to oral glucose challenge in obese people with type 2 diabetes either alone or in combination with sitagliptin, a DDP-4 inhibitor, and proposed a triple acting role of dorzagliatin in regulation of glucose homeostasis. I have no further comments.